# From Lignocellulosic Residues to Protein Sources: Insights into Biomass Pre-Treatments and Conversion

**DOI:** 10.3390/polym17162251

**Published:** 2025-08-20

**Authors:** Isabela Vera dos Anjos, Natacha Coelho, Hugo Duarte, Diogo Neves Proença, Maria F. Duarte, Raul Barros, Sara Raposo, Sandra Gonçalves, Anabela Romano, Bruno Medronho

**Affiliations:** 1MED—Mediterranean Institute for Agriculture, Environment and Development, CHANGE—Global Change and Sustainability Institute, Faculdade de Ciências e Tecnologia, Universidade do Algarve, Campus de Gambelas, Ed. 8, 8005-139 Faro, Portugal; ivanjos@ualg.pt (I.V.d.A.); natacha.coelho@necton.pt (N.C.); hmduarte@ualg.pt (H.D.); daproenca@ualg.pt (D.N.P.); smgoncalves@ualg.pt (S.G.); aromano@ualg.pt (A.R.); 2Necton S.A., Belamandil, 8700-152 Olhão, Portugal; 3University of Coimbra, Chemical Engineering and Renewable Resources for Sustainability (CERES), Department of Chemical Engineering, Rua Sílvio Lima, Pólo II, Pinhal de Marrocos, 3030-790 Coimbra, Portugal; 4University of Coimbra, Centre for Mechanical Engineering, Materials and Processes (CEMMPRE), Advanced Production and Intelligent Systems (ARISE), Department of Life Sciences, Calçada Martim de Freitas, 3000-456 Coimbra, Portugal; 5MED—Mediterranean Institute for Agriculture, Environment and Development, CHANGE—Global Change and Sustainability Institute, CEBAL, 7801-908 Beja, Portugal; fatima.duarte@cebal.pt; 6CIMA—Centre for Marine and Environmental Research, ARNET—Aquatic Research Networks, Faculdade de Ciências e Tecnologia, Campus de Gambelas, Universidade do Algarve, 8005-139 Faro, Portugal; rbarros@ualg.pt (R.B.); sraposo@ualg.pt (S.R.); 7Surface and Colloid Engineering, FSCN Research Centre, Mid Sweden University, SE-851 70 Sundsvall, Sweden

**Keywords:** single-cell protein, bioconversion, circular economy, renewable resources, microbial fermentation

## Abstract

With the global population steadily rising, the demand for sustainable protein sources has become increasingly urgent. Traditional animal- and plant-based proteins face challenges related to scalability, resource efficiency, and environmental impact. In this context, single-cell protein has emerged as a promising alternative. Derived from microorganisms such as algae, bacteria, fungi, and yeast, single-cell protein offers a high nutritional profile- including all essential amino acids and vitamins—while enabling rapid production, minimal land and water requirements, and no generation of greenhouse gas emissions. A particularly compelling advantage of single-cell protein is its ability to be produced from agro-industrial waste, converting low-cost residues into valuable nutritional resources and contributing to environmental sustainability. Among these waste streams, lignocellulosic biomass from agricultural and forestry residues stands out as a renewable, biodegradable, and abundant feedstock. This review explores the potential of lignocellulosic waste as a substrate for single-cell protein production, emphasizing both its environmental advantages and nutritional value. It highlights the single-cell protein role as a sustainable and scalable alternative to conventional protein sources. The review also identifies key scientific, economic, and regulatory challenges, and recognizes the importance of targeted investments, particularly in policy development, public awareness, and technological innovation, to enable the broader adoption and acceptance of single-cell protein -based products.

## 1. Introduction

Proteins are the main structural constituents of our cells and tissues, are required for metabolic function and are crucial for the growth and development of our body. Proteins are composed of amino acids; while some non-essential amino acids can be synthesized by our body, others—known as essential amino acids—must be obtained through the diet. Since proteins are continuously repaired and replaced throughout our lives, our bodies must consistently obtain amino acids, making proteins an essential component of the human diet.

One key source of amino acids is milk proteins, including whey and casein, which have gained significant popularity in the food and supplement industries. These proteins are highly valued due to their high biological value, ease of digestion, and comprehensive profile of essential amino acids. Whey protein, a by-product of cheese and yoghurt production, is rapidly absorbed and is rich in branched-chain amino acids, especially leucine, which makes it effective in promoting muscle protein synthesis. On the other hand, casein is characterised by a slower digestion process, resulting in the formation of a gastric gel that facilitates the sustained release of amino acids. This property is particularly advantageous in preserving muscle mass during periods of fasting [1,2]. However, despite the nutritional benefits, the production of these proteins is associated with significant environmental impacts. These include high water consumption, greenhouse gas emissions, and the generation of potentially polluting waste, particularly when whey is not utilised effectively [3].

The global population is predicted to increase to approximately 9 billion by 2050 and 11 billion by 2100, stressing the urgent need to find sustainable solutions able to secure essential resources, especially arable land, freshwater, and food. Traditionally, food production systems rely on animal-based protein sources. However, these systems are increasingly unsustainable and well-known contributors to environmental degradation [4]. In addition, if the current level of protein consumption is maintained, the global average demand for animal-sourced food products is estimated to increase from 1.4 billion tons to 2.0 billion tons in 2050 [5]. Moreover, while nowadays approximately five times more protein (animal and plant) is produced than is required to feed the world, only 34% of the produced protein is directly consumed by humans, and more than 800 million people are malnourished [6,7].

Though in the past few decades, the alternative protein industry has focused on high-income countries, low- and middle-income countries offer potential for market growth as well. Overall, driven by increasing awareness of the ethical and environmental implications of livestock production, the demand for alternative protein varies according to social, cultural, economic, and environmental factors [8]. Therefore, there is an urgent need for protein alternatives that align with current consumer demands, including plant-based, insect-based, cell-based/cultivated meat, and microbial proteins. In the latter category, microbial protein or single-cell protein (SCP) is the term that defines dried biomass or proteins extracted from algae, bacteria, filamentous fungi, or yeast cell cultures [9,10,11]. Due to its high protein content (ranging from 60 to 82% of dry cell weight) and rich amino acid composition, including the essential amino acids lysine and methionine, SCP is used as a dietary supplement in human and animal feed [9,10,12,13]. Moreover, SCP also contains vitamins, minerals, carbohydrates, and lipids, providing an additional nutritional value [9,12,13].

Compared to conventional protein sources, SCP production still lacks investment to overcome its major limitations, such as substrate availability, process scalability, monitoring, and process control [14]. Nonetheless, SCP offers several advantages, such as rapid growth in limited space, independence from climate variations, low water requirements, and zero greenhouse gas emissions. Its ability to be produced from agro-industrial waste makes SCP an attractive, renewable, and sustainable protein source [11,15,16]. From a circular economic perspective, using waste as a substrate for SCP production not only addresses waste management challenges but also reduces environmental pollution and production costs by providing inexpensive raw materials. An ideal substrate should be non-toxic, regenerative, cost-effective, and widely available to ensure economically viable and high-quality SCP production. The key to efficient production of SCP largely depends on optimizing substrate bioconversion, which facilitates rapid microbial growth [9,17].

Lignocellulosic biomass is known to contain more than 60% of fermentable sugars. However, its recalcitrance and high crystallinity make it difficult to extract sugars from the plant cell wall. In general, lignocellulosic biomass is composed of cellulose, hemicellulose and lignin, among a variety of biopolymers, depending on its source [18]. Hemicellulose is more easily hydrolyzed than cellulose, and besides yielding glucose, it can be degraded into several sugars, such as mannose, galactose or xylan. On the other hand, the hydrolysis of lignin (a non-sugar polymer) originates compounds that hinder the degradation of the other constituents of the biomass [19]. This biomass can be obtained from waste generated from agricultural and forestry practices, generated in large quantities worldwide, with a substantial part often discarded and/or incinerated [12]. However, the unique properties of lignocellulosic biomass endow it with tremendous biotechnological potential, as it can be converted into a wide range of value-added products, including biofuels, biofertilizers, biochemicals, and biopesticides. Overall, lignocellulosic biomass is a valuable and abundant carbon source, readily available and accessible as feedstock for microbial bioconversion processes, not competing with resources used for human food production. Given its relevance and potential, this review focuses on recent advances in the use of lignocellulosic biomass for SCP production over the past few years [15,20].

## 2. Lignocellulosic Biomass

### 2.1. Composition and Structure of Lignocellulosic Biomass

As stated in the report *Mapping and Synthesis of International Biomass Supply Assessments*, published in January 2025 by Oak Ridge National Laboratory (ORNL) for the United States Department of Energy (DOE), the initial projection of sustainable biomass supply from 62 countries exceeds 2.8 billion tonnes. However, it should be noted that this figure represents only a fraction of the global potential, as the criteria used to quantify sustainable supplies varied significantly among national assessments. Brazil, the United States, China, Indonesia, and India are among the leading contributors to this estimate [21]. Complementary studies focused specifically on lignocellulosic biomass estimate that global annual production surpasses 180 billion tonnes [22,23,24]. Although this figure predates the ORNL report, it underscores the magnitude and availability of this renewable resource on a global scale.

Lignocellulosic biomass, or simply lignocellulose, is the most abundant renewable organic material on Earth. It forms the primary structural framework of plants, providing mechanical strength through a complex matrix, primarily composed of cellulose, hemicellulose, and lignin (Figure 1), which together account for approximately 90% of its dry weight. The remaining 10% consists of ash and extractives. The complex matrix is characterized by intricate covalent and non-covalent interactions, where cellulose microfibrils are embedded within a heterogeneous network of hemicellulose and lignin [20,25].

The composition and proportions of lignocellulosic components vary across plant species and are also influenced by environmental factors. Regardless, cellulose remains the dominant component of lignocellulose (30–60%) and is the world’s foremost organic polymer and renewable carbon source [20,25]. Chemically, cellulose (C_6_H_10_O_5_)_n_ is a water-insoluble polysaccharide composed of linear chains of *β*-D-glucose units connected by *β*-1,4-glycosidic bonds. These chains are arranged into microfibrils, held together by hydrogen bonds and van der Waals forces, forming cellulose fibers. This organization creates ordered crystalline regions that are highly resistant to degradation, as well as less resilient amorphous (disordered) regions [15,20,25,26,27].

In plants, cellulose forms the structural backbone of cell walls, which are covered by hemicelluloses and lignin, filling gaps between microfibrils and providing additional stability to the cell wall structure. Hemicellulose, a branched heteropolymer consisting of different sugar units arranged in a complex structure, is the second major constituent of lignocellulose (20–40%) [15,26]. Unlike cellulose, which is only composed of glucose units, hemicellulose consists of pentoses (D-xylose and L-arabinose), hexoses (D-glucose, D-galactose, D-mannose, L-rhamnose, L-fucose) and acetylated sugars [15,25,26]. The content of sugar units in hemicellulose varies according to plant species, development stage, and tissue type, with most of the main-chain sugars bonded by β-1,4-glycosidic linkages. In hardwood species, hemicellulose is predominantly composed of xylan, while glucomannan is the major component in softwood species. Due to its lower degree of polymerization (500–3000 sugar units) compared to cellulose (7000–15,000 glucose units), hemicellulose forms amorphous structures rather than crystalline ones, making it easier to degrade [26,28,29]. In nature, hemicellulose cross-links with cellulose fibers and lignin through extensive hydrogen bonding and cinnamic acid ester linkages, respectively, providing additional firmness to the plant cell wall. Lignin is the third major constituent of lignocellulose, an aromatic polymer of phenylpropane units, such as p-coumaric alcohol, coniferyl alcohol, and sinapyl alcohol, cross-linked to form a large three-dimensional network structure. Although lignin is not a source of fermentable carbon, its amorphous nature provides impermeability and resistance against microbial degradation and oxidative damage within the cell wall [20,25,28,29].
Figure 1Chemical structures of cellulose, hemicellulose and lignin. Adapted from Khaire et al. [29].
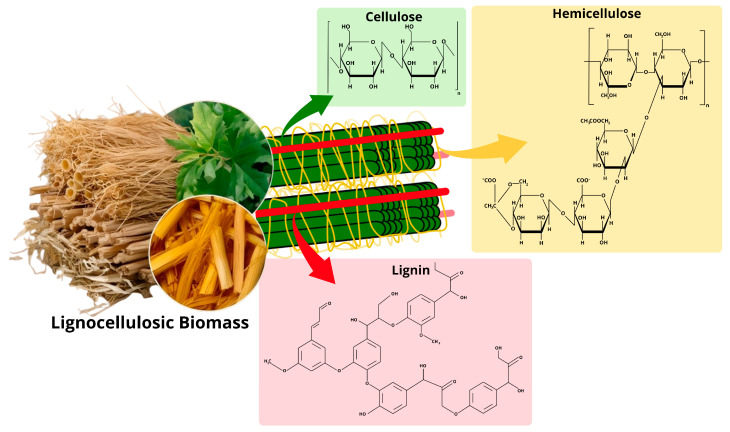



### 2.2. Bioconversion of Lignocellulosic Biomass

#### 2.2.1. Lignocellulose Pre-Treatments

As a major source of renewable organic matter and the primary structural component of plants, lignocellulosic biomass serves as a readily available raw material for microbial bioconversion and, more importantly, its utilization does not compete with raw materials used for human food production [15,30]. The conversion of lignocellulosic biomass into SCP offers a sustainable strategy for producing reliable protein supplements for both human food and animal feed. However, inherent lignocellulosic complexity limits the conversion to only about 20% of the original cellulose into fermentable sugars [31,32]. In addition to these components, lignocellulosic biomass also contains proteins, lipids, pectin, minerals, and other compounds that contribute to its recalcitrance. Therefore, pre-treatments are essential to dismantle the lignocellulose complex hierarchical structure prior to bioconversion. These pre-treatments increase the availability of carbohydrates and other valuable molecules present in biomass, thereby maximizing yield [15,31,32,33]. In this regard, various physicochemical methods can be employed within an integrated and synergistic framework to minimize carbohydrate loss, achieve maximum delignification, and consequently enhance the overall efficiency of the biorefinery process [34].

In general, pre-treatments involve the use of two or more methods to minimize the adverse conditions often encountered in single-step approaches, thereby preventing the loss of valuable compounds and reducing the generation of microbial growth inhibitors. Moreover, an optimal sequence of pre-treatments is essential, as it significantly affects cellulose accessibility to enzymes. Typically, most integrated pre-treatments target the removal of amorphous hemicellulose in the initial step, followed by subsequent steps aiming at lignin solubilization. Commonly used in combination, pre-treatment methods can be categorized as physical, physicochemical, or biological, depending on their specificities (Figure 2) [34,35].

Physical pre-treatments include a variety of techniques, such as ultrasound, microwaves, extrusion, milling, and grinding, among others [32,36]. Though quite energy consuming and representing up to 20% of operation cost, these methods are fundamental to reduce the biomass particle size, increasing the available surface area for chemical, microbial or enzymatic reactions. Although requiring specialized equipment, physical pre-treatments remain among the most effective and straightforward methods to break down plant biomass as raw material. On an industrial level, to overcome its limitations in fully degrading biomass, some physical pre-treatments are typically combined with the addition of chemical pre-treatments [37,38].

Chemical pre-treatment methods involve using chemical agents such as alkali, acids or organic solvents to degrade biomass [32]. Among these, dilute acid solutions combined with high temperature and pressure are frequently employed. Despite its highly efficient biomass degradation, concentrations higher than 10% (*v*/*v*) are usually avoided to prevent equipment corrosion, carbohydrate loss and production of biological inhibitors like phenols, organic acids and aromatic aldehydes [32,39].

Both physical and chemical characteristics of biomass can be altered through a hybrid pre-treatment approach, which can be defined as physicochemical. This approach, typically involves the combination of high temperature and pressure with a chemical process, to create harsh conditions able to break lignocellulosic biomass. One of the most studied physicochemical pre-treatments is steam explosion. This method involves exposing lignocellulosic biomass to a combination of high-temperature pressurized steam and subsequent pressure release, forcing steam into the porous network of the substrate, which rapidly expands upon decompression, inducing evaporation of condensed moisture and disaggregation of the biomass matrix. This process alters the chemical structure of the substrate by disrupting intermolecular interactions, such as hydrogen bonds, ultimately leading to hydrolysis. Other physicochemical pre-treatments like ammonia fiber explosion or CO_2_ explosion are based on the same concept while varying the active chemical [39,40]. The application of electromagnetic radiation in the form of microwave irradiation induces comparable effects. The increase in pressure and temperature causes the rupture of cell walls, disrupting the lignocellulose structure while reducing particle size and increasing outer surface area. This effect can be further increased with the addition of NaOH, though requiring subsequent neutralization [41]. However, minimizing the presence of enzymatic and yeast fermentation inhibitors is always desirable. As an alternative, hydrothermal methods such as liquid hot water have demonstrated high efficiency with low environmental impact [42]. Though mainly using water, inhibitors as furfural, formic, acetic and levulinic acids can still be formed during biomass hydrolysis through liquid hot water or steam explosion pre-treatments [43]. The selection of an appropriate pre-treatment method depends on the functionality and mechanism of the process. For example, hot water pre-treatment is more suitable for biomass with low lignin content, whereas acid or alkaline pre-treatments are adequate for biomass with high lignin content. To overcome these limitations, Gundupalli et al. [44] demonstrated that combining hot water and a deep eutectic solvent in a two-stage pre-treatment improved hemicellulose recovery and delignification from a solid mixture of sugarcane bagasse, rice straw, and napier grass. In the pursuit of more environmentally friendly pre-treatment alternatives, ultrasound technology has emerged as a promising technology [45]. The application of ultrasound waves has shown to overcome the recalcitrance of lignocellulosic biomass, not only on its own but also in combination with other methods, improving their efficiency. Ultrasound waves generate acoustic cavitation, initiating the spontaneous formation of microbubbles, which, upon collapsing, release high amounts of energy, leading to high temperature and pressure in the surrounding area [45,46]. This effect has proven to be a critical factor in improving the biomass conversion efficiency at the industrial scale. Recently, Cabrera et al. [47] proposed an innovative mechanical design for a continuous ultrasound bath focused on the pre-treatment of peapod biomass. A continuous step is proposed, with the aim of reducing processing time while improving efficiency. Moreover, sonochemical treatment based on the application of ultrasounds under alkali conditions has already proved its effectiveness as an optimal method to make biomass more susceptible to enzymatic hydrolysis, thereby increasing the yield of fermentable sugars [48].

The final stage of pre-treatment often involves the application of biological methods, utilizing the activity of enzymes, fungi or bacteria [49,50]. Biological pre-treatments are mostly focused on enzymatic activity aiming at reducing cellulose degree of polymerization, hydrolyzing hemicellulose and removing lignin. The enzymatic efficiency on lignocellulose biomass is largely influenced by the type of enzyme used, biomass features, and its plant-specific composition. Nonetheless, parameters as enzyme concentration, interaction between enzymes and biomass components, and reaction time should not be neglected [51,52]. Although enzymes can also be obtained from plants and animals, commercial enzymes are obtained from bacteria or fungi due to their suitability for large-scale production. In this context, genetically engineered strains or bacterial isolates can be used to produce enzymes to target a specific substrate, such as cellulose, hemicellulose or lignin [51,53,54]. To ensure successful and efficient pre-treatment, each one of these polymers requires the action of specific enzyme groups. As cellulases hydrolyze the β-(1, 4) linkages present in cellulose, hemicellulases are capable of hydrolyzing the xylan from hemicellulose, while ligninases degrade non-phenolic lignin units mainly by peroxide catalysis [55,56,57]. The selection of bacterial strains is influenced by the characteristics of the substrate. Cellulolytic bacteria have been explored for the degradation of lignocellulose also due to their resilience, with some capable of enduring harsh conditions and remaining viable in wide ranges of pH, temperature and salinity [58]. Though not as effective, using a microbial population can be more versatile than an enzymatic approach, as bacteria can synthesize enzymes with different activities. For instance, a cellulolytic strain of *Serratia marcescens* was found to effectively saccharify bamboo biomass. Peaks of enzyme activity were found to differ in type of enzyme and fermentation time, with most lignin and hemicellulose degrading enzymes reaching their peak activity on the third fermentation day, while cellulose degrading enzymes reached their maximum activity on the sixth day [59]. To further overcome its recalcitrance and improve the degradation of lignocellulosic biomass, microbial communities present a promising strategy. This approach, considered a consolidated bioprocess, is also based on the synergistic activities of multiple secreted enzymes, usually by bacterial or fungal communities [60,61].

Though requiring longer fermentation periods, fungi have been explored for the degradation of lignocellulosic biomass and enzyme production. Moreover, the fungal pre-treatment of lignocellulose has the advantages of usually not needing chemical additives, along with low energy input and operational costs [62,63]. Several fungi, including brown rot fungi, white and soft rot fungi, are used for lignocellulose degradation [50]. Though their degree of delignification varies greatly depending on the fungi species and carbohydrate content in biomass; white-rot fungi have been reported as the most effective decomposers of lignin. Some species selectively degrade lignin, and others do not. White-rot fungi produce extracellular oxidative enzymes (mainly laccase and lignin peroxidase) that function as biodegradation agents, breaking down lignocellulose into simpler molecules [64,65]. Fungal consortia have proven their increased effectiveness in lignocellulose degradation. Sajid et al. [66] showed significant differences in the lignocellulolytic enzyme production between five fungal isolates and how rice straw biomass degradation can be enhanced when using a fungal consortium pre-treatment. A mixture of rot-fungal strains was also used as an alternative process to minimize the generated cardboard waste. When pre-treated with a fungi consortium, the cardboard waste showed a rapid mineralization and a rapid decrease in its total organic carbon, carbon/nitrogen ratio and lignocellulose content [67]. Overall, fungal pre-treatment of biomass is mainly focused on delignification to further increase the accessibility of hemicellulose and cellulose for subsequent enzymatic hydrolysis. Furthermore, integrating biological pre-treatments with milder physicochemical methods has the potential to develop lignocellulose bioprocessing towards sustainability, strengthening the green economy by providing a sustainable and renewable source of bioproducts [68].

As mentioned, since different types of lignocellulosic biomass are available, no single pre-treatment method can efficiently convert all types into free sugars. An ideal method would eliminate the need for biomass size reduction, avoid the generation of toxic inhibitors, and operate with minimal energy inputs at the lowest cost. The selection of an appropriate pre-treatment should thus also consider the environmental compatibility of the chemicals or methods used in the entire process. For instance, in industrial production, the feasibility of a method is judged not only by its effectiveness but also by the overall energy consumption and operational costs [36,69,70]. In this context, it is essential to consider not only the technical efficiency of each pre-treatment method, but also its energy demand and economic feasibility. Physical methods such as milling, microwave irradiation, and ultrasound have been shown to be effective in reducing particle size and increasing surface area. However, these methods are often energy-intensive, which can pose challenges at an industrial scale [71].

The utilisation of chemical methodologies, encompassing dilute acid or alkali treatments, is similarly constrained by factors pertaining to energy demand, reagent corrosiveness, and the generation of microbial inhibitors. Despite the efficacy of these methodologies in disrupting the lignocellulosic structure, their large-scale implementation necessitates meticulous consideration of sustainability and integration with other process stages [72].

Consequently, hybrid and integrated strategies have gained prominence. The integration of physicochemical methodologies with biological approaches has demonstrated potential in reducing costs, enhancing efficiency, and minimising environmental impacts. It is important to recognize that no singular methodology is optimal in all circumstances; rather, the selection of a method should be informed by considerations such as biomass composition, resource availability, and the overarching objectives of the bioprocess [73].

To complement the discussion, Appendix A presents a comparative overview of the main pre-treatment methods applied to lignocellulosic biomass. The table summarizes typical sugar yields, process durations, and relative costs based on data from the literature. However, it is important to note that quantitative comparisons across pre-treatment methods are inherently limited, as performance metrics such as sugar release, inhibitor formation, and energy demand are strongly influenced by the type and composition of the biomass [32,43,51]. For instance, dilute acid pre-treatment may be highly effective for agricultural residues but less suitable for woody biomass due to increased inhibitor formation [39,43]. Likewise, enzymatic efficiency varies depending on substrate structure, enzyme specificity, and microbial source [51,52,53]. Therefore, while the table provides a general framework for comparison, the selection of an appropriate pre-treatment strategy must be tailored to the specific biomass and process objectives, considering technical, economic, and environmental factors [74].

Finally, it is worth noticing that fixed or universal cost estimates are not feasible for pre-treatment methods. The financial outlay required is subject to considerable variation, depending on the type of biomass, pre-treatment chemicals, microbial strains, process scale, and operational conditions [74]. Consequently, the necessity arises for feasibility assessments to be conducted on a case-by-case basis, with due consideration given to technical, economic, and environmental criteria. Overall, the most adequate and efficient procedure for the pre-treatment of lignocellulosic biomass, based on current knowledge and available technology, should start with biomass physicochemical hydrolysis, followed by a biological pre-treatment.
Figure 2Various pre-treatment techniques for lignocellulosic biomass. Adapted from Hashemi et al. [70].
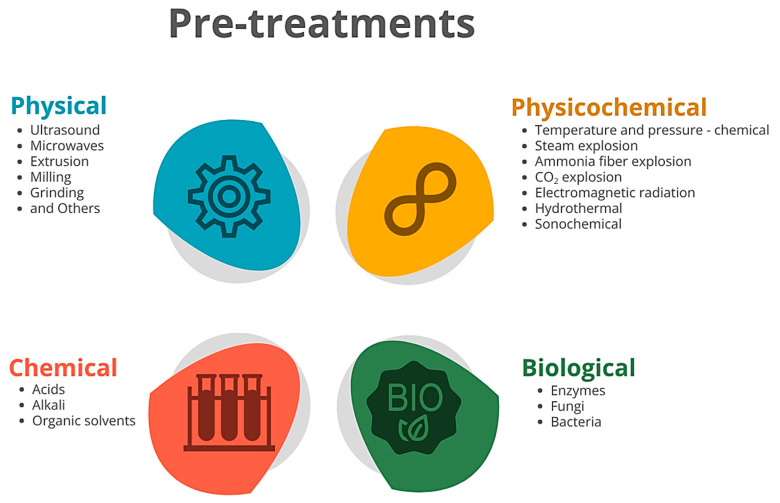



#### 2.2.2. SCP Production Methods

After pre-treatment, fermentation begins, involving the bioconversion of the substrate into the desired product [75]. The selection of an adequate fermentation technique must consider the microorganism and, especially, the substrate, as each species has unique nutrient requirements [76]. Additionally, environmental protection and safety concerns must also be considered [77]. SCP production offers environmental benefits in terms of land use and water efficiency, adding to lower energy requirements and greenhouse gas emissions in comparison with animal-based sources. However, the extent of the environmental benefits associated with SCP production should consider the safety and environmental impact of effluents from chemical pre-treatments, as well as the substitution levels of electricity usage from fossil fuels for renewable sources [14]. In addition, the underlying technology should minimize the use of complicated purification methods, finding biotechnological alternatives [14,78].

In SCP production, fermentation methods can be classified into submerged, semi-solid, and solid-state types [15,77]. In liquid submerged fermentation (LSF), microorganisms are cultivated in a free-flowing liquid medium where the carbon source is dissolved or suspended [18,79]. This substrate would need to be pre-treated to release fermentable sugars that can be dissolved or suspended in the liquid medium. This process involves microbial cultivation in a continuous liquid-phase substrate containing over 95% water. It is conducted in a closed bioreactor, typically in continuous mode, with precise control over temperature, pH, nutrients, and oxygen supply [14]. Additionally, aeration is generally required due to the potentially high cell densities achieved on the metabolized sugars from lignocellulosic biomass, leading to greater oxygen demands [76]. After fermentation, the resulting biomass, constituting the SCP, is recovered through centrifugation or filtration and drying [77]. Although purification is easier with submerged fermentation, this method faces significant challenges, including high operational costs and substantial technical investment [15,79]. According to Ribeiro et al. [80], the most commonly used microorganisms in submerged fermentation include strains belonging to the genus *Saccharomyces*, followed by strains from the genera *Candida*, *Kluyveromyces*, and *Aspergillus*.

Semi-solid fermentation involves an intermediary approach between solid-state fermentation (SSF) and LSF. In this approach, a pre-treated lignocellulosic substrate retains some free liquid, which enhances nutrient availability and allows better control over the fermentation process [81]. Key steps include agitation, mixing in a multiphase system, oxygen transport, and heat removal [77]. This method is cost-effective, as it closely mimics the natural environment of many microorganisms and typically utilizes low-cost agricultural residues [15]. The selection of an adequate bioreactor for fermentation is crucial and should be based on the productivity and specific requirements for the growth of the selected microorganism [82]. In this regard, the control of different key variables, including substrate concentration, oxygen levels, and the presence of toxic inhibitory compounds, should be strikingly controlled [83].

SSF, on the other hand, involves an insoluble substrate with no free liquid present [18,83]. In SSF, microorganisms obtain nutrients by adsorbing or penetrating the solid substrate, allowing them to proliferate and generate protein-rich biomass. A nutrient concentration gradient naturally develops in SSF, making diffusion essential for optimal microbial growth. Additionally, an adequate oxygen supply in the liquid phase is crucial to maintain fermentation efficiency. This is ensured through aeration and intermittent stirring to facilitate proper gas exchange.

Maintaining optimal parameters, including temperature, pH, ionic strength, and nutrient availability, is vital to maximizing yield and overall process efficiency. This technique, though traditionally used for products like bread and cheese [84], has recently gained attention for producing enzymes, foods, SCP, and more, due to its economic feasibility and low energy demands [15,18]. Additionally, SSF generally offers higher protein yields compared to submerged fermentation, especially for fungal growth [84]. This technique is particularly suitable for microorganisms well-suited for growth under low water activity, such as yeast and certain filamentous fungi, that prefer pure solid substrates with ~60–65% moisture content [14]. The relatively low capital investment and minimal waste generation associated with SSF make it an environmentally and economically attractive option for SCP production from lignocellulosic biomass [80]. After SCP production, one of the main cost-related challenges is protein harvesting. Typically, even with high yield production levels, the resulting solutions are diluted, needing the implementation of additional methods to concentrate the protein, such as filtration, precipitation, centrifugation, or the use of semi-permeable membranes [15]. Table 1 summarises the main advantages and disadvantages discussed in this section.

### 2.3. Microorganisms Used in Single-Cell Protein Production

A diversity of microorganisms can be used in the bioconversion of lignocellulosic biomass into SCP. Key requirements for SCP production include the ability to utilize a diverse range of low-cost carbon and nitrogen sources and the moderate growth conditions necessary for converting the substrate into a valuable product, which leads to rapid growth and high productivity [15,16]. Various types of substrates and microorganisms used in recent years are summarized in the Appendix A. This is highly relevant from a circular economy perspective, as finite resources can be complemented by using waste/by-products as carbon-nitrogen sources to produce value-added products.

In recent studies, molecular identification of SCP-producing strains has been increasingly adopted to ensure accurate taxonomic classification and functional characterization. Techniques such as D1/D2 domain of the 26S rRNA gene and 28S rRNA gene (LSU) sequencing have been used to identify filamentous fungi and yeasts, including *Aspergillus niger*, *Penicillium citrinum*, and *Candida intermedia* [85]. These methods allow for precise strain selection and monitoring during fermentation processes, especially when working with environmental isolates or non-conventional strains.

When selecting a microorganism for SCP production, several criteria must be considered: tolerance to pH, temperature, and salinity, oxygen dependence; lack of pathogenicity or toxicity to humans, animals or plants; and the nutritional quality of the biomass produced. The microorganism should be readily harvestable, provide high protein yield and enable low production costs [15,16,76].

Microorganisms used for SCP production are classified into four main categories: algae, bacteria, fungi, and yeasts (Figure 3) [86]. While fungi are the mostly used microorganisms in SCP production studies, algae are simple to cultivate and efficiently convert solar energy, producing SCP with high protein content (≥70%) along with essential nutrients, such as omega-3 fatty acids, minerals, and vitamins. However, algae cultivation requires sunlight, which can pose a disadvantage in algae usage. Also, algae require warm temperatures and CO_2_, and their cell walls are difficult to digest [12,13,16,87].

Bacteria have a higher protein content (50–80% of dry weight) and can grow on a wide range of materials with a rapid growth rate, as is the case with *Methylophilus* spp. (within 120 min) [14,87]. However, their small cell size and low density negatively affect biomass harvesting. Additionally, the high nucleic acid content and the limited public acceptance of bacteria as a food source are limiting factors in their use for SCP production [12,15,16]. The bacterial-based production process tends to be more costly due to the need for additional steps, such as the removal of certain amino acids, after fermentation. Ongoing research seeks to optimize this process, aiming to increase yield and improve the economic viability of using bacteria for SCP production [87].

Among the microorganisms used in SCP production, fungi have gained prominence due to their increasing acceptance and advantages, including larger size, higher nutritional value, and ease of harvesting [88,89]. In addition to protein and vitamins, fungi typically contain a moderate nucleic acid content of around 10%, which is considered unsuitable for human consumption and thus requires processing to decrease it [90,91,92]. Traditionally, various fungal species are used in the production of a wide range of fermented foods and beverages [93]. Many well-known food products result from traditional fermentation methods involving fungi and yeasts, such as soy sauce, miso, moldy cheeses, beers, wines, and spirits. Edible mushrooms (the fruiting bodies of macrofungi) are also widely recognized as food valued for their high nutritional content, being rich in proteins, vitamins, and antioxidants [83,94]. Filamentous fungi are commonly used due to their advantage in harvestability owing to their larger size, although they have a slower growth rate and lower protein content [79,86]. Studies on SCP production with various species of fungi are ongoing, leading to the development of innovative products and production processes [83].

Yeasts are also widely used, and the protein content in their biomass is approximately 55 to 60%, though these values may vary depending on cultivation conditions (medium, temperature, and time) and the strain used [12,95,96]. Although yeasts have a lower growth rate and lower protein and methionine content compared to bacteria, yeast SCP provides balanced and high-quality protein in terms of amino acids, suitable for both human and animal consumption. When compared to traditional crops or livestock, yeast cultivation and harvest are cheaper, easier and faster. Given its adaptability, yeast can grow in a variety of substrates, comprising a wide range of hydrophilic and hydrophobic wastes and by-products, biomass and raw materials. In particular, a yeast protein content above 50% has been reported from substrates such as prawn-shell waste, rice straw hydrolysate, mango waste, vinasse, olive fruit wastes or papaya peel [95,97].

Several SCP-relevant microorganisms have had their genomes sequenced, enabling deeper insights into metabolic pathways and optimization strategies. For example, *Yarrowia lipolytica* strain A101 has a publicly available draft genome (GCA_900518095.1) [98] and has been successfully used in SCP production from lignocellulosic substrates, such as rye and oat bran hydrolysates [99]. Transcriptomic studies have also explored its metabolic potential under different substrate conditions [100]. Other species, such as *Cyberlindnera jadinii* CBS 5713 (GCA_000006775.2) and *Wickerhamomyces anomalus* NRRL Y-366-8 (GCA_002083305.1), also have genome assemblies available and were successfully used in SCP production from lignocellulosic residues [101].

In recent years, several studies have investigated the production of SCP using waste as a substrate, making use of several types of microorganisms as summarized in Table 1. However, only a limited number of studies have specifically addressed the use of lignocellulosic waste. Among these, most have utilized algae and yeasts as microorganisms [85,99,101,102,103,104,105,106].
Figure 3Principal microorganisms used in SCP production and their main characteristics. Adapted from Pereira et al. [106].
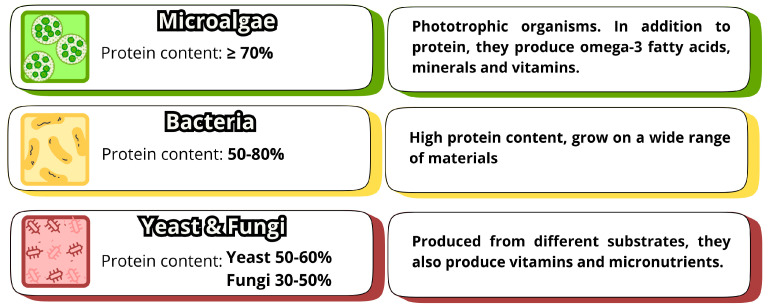



### 2.4. Nutritional Benefits of SCP

Proteins are vital macromolecules involved in cellular structure and metabolic functions. Thus, the growth of the global population combined with environmental challenges has significantly increased the demand for protein-rich foods, inducing considerable pressure on global food systems. In this context, SCP is gaining prominence as a promising alternative to address global food shortages, although increasing public familiarity with this protein source is essential. In addition to meeting safety requirements, efforts are needed to enhance acceptance by raising public awareness about the benefits of incorporating SCP into the human diet [12]. Interest in SCP production is growing in parallel with advances in its development. Research highlights a range of potential physiological benefits associated with its consumption, although the specific amounts of compounds can vary depending on the method, substrate, and especially the microorganism used for production. While SCP is most commonly produced by fungi, other microorganisms like yeast, bacteria, and, more recently, microalgae have also shown relevant nutritional characteristics (Table 2) [12,107].
polymers-17-02251-t002_Table 2Table 2Single-Cell Protein sources and their nutritional quality.Type of SCPProtein Content (% Dry Weight)Essential Amino AcidsOther NutrientsBenefitsReferences**Fungal**30–50%High in lysine and threonine, complete essential amino acid profile (including methionine)Fibres (chitin, β-glucans), unsaturated fatty acids, vitamins B9, B12, choline, minerals (Ca, Mg, Zn)Reduces LDL cholesterol, increases satiety, lowers glycaemic responseRajput et al. [86]; Finnigan et al. [107]; Wiedeman et al. [108]; Derbyshire & Delange [109]; Nyyssölä et al. [110]; Turnbull et al. [111]**Yeast**40–55%Contains all essential amino acidsB-complex vitamins, minerals, fibres, antioxidantsAntioxidant potential, functional food, food additiveJach et al. [95]; Timira et al. [112]; Saravanan et al. [113]**Bacterial**50–80%Variable, rich in histidine, valine, and leucineCarbohydrates, fibres, fats, high nucleic acid content (may limit direct human use in large quantities, but can be reduced by processing)Grows on diverse substratesBratosin et al. [12]; Sharif et al. [16]; Kaur & Chavan [114]; Zha et al. [115]; Hadi & Brightwell [116]**Microalgae**30–80%Complete, species-dependentVitamins, minerals, fibres, low nucleic acid contentImmune support, cardiovascular and chronic disease preventionJanssen et al. [117]; Montenegro Herrera et al. [118]; Grosshagaue et al. [119]; Koyande et al. [120]; Geada et al. [121]**Mixed**40–70%Balanced, depending on the combinationCombined nutrients from various sourcesOptimised nutritional profile, cost-effective productionWang et al. [122]; Rasouli et al. [123]; Hülsen et al. [124]


Fungal-derived SCP has received greater attention due to its nutritional profile, with its protein content able to reach 30–50% of the fungi’s dry weight, under ideal production conditions [86]. This type of protein, known as mycoprotein, is considered of high quality, and is characterized by low energy content, high fiber content (chitin, *β*-glucans), as well as fats (unsaturated fatty acids, linoleic and linolenic acids), carbohydrates, and minerals [107]. Regarding micronutrients, fungal SCP is rich in vitamins B9 and B12, calcium, phosphorus, magnesium, and zinc. It also contains higher levels of choline compared to other known sources like salmon, pork, soy, bacon, and wheat germ [108,109]. Although fungal SCP generally has low methionine content, it typically has high levels of threonine and lysine, thus meeting the minimum requirements set by the Food and Agriculture Organization (FAO) [86,92,110].

Fungal SCPs contain higher levels of lysine than those SCPs produced by bacteria or algae, making them a nutritionally valuable alternative, provided that their nucleic acid levels and potential mycotoxins are reduced or completely removed when necessary [125]. In addition to its high content in proteins and fibers and low in carbohydrates, fungal SCP has also demonstrated effectiveness in reducing total cholesterol and LDL, the so-called “bad cholesterol”. According to Turnbull et al. [111], this reduction may be attributed to a specific fiber profile that likely influences the intestinal microbiota, causing cholesterol reduction. There are also indications that fungal SCP can increase satiety and reduce the glycemic response when consumed with carbohydrates [107]. Yeast has been predominant in the culinary world for centuries and is recognized for its rich content of vitamins, minerals, fiber and proteins. However, its nutrient load and composition of cellular components significantly vary with the distinct growth phases, leading to differences in nutrient profiles. Altogether, yeast is considered a source of high-quality protein as it contains all the essential amino acids required for human nutrition [112]. It also has notable antioxidant properties and potential to be used as a dietary additive with the capacity to positively impact human health [95,113]. Although less commonly used, some studies have shown that bacterial-derived SCP is also rich in carbohydrates, fats, and fibers, with a protein content ranging from 50 to 80% of dry weight [12,114]. Similar to yeast, bacteria can grow in a large variety of raw materials and edible substrates, easily multiply on waste of organic matter and in water-based media supplemented with minerals and nutrients. The selection of a specific strain usually depends on the available substrate. For example, wheat bran is adequate for *Rhodocyclus gelatinosus* growth, potato waste for *Bacillus liqueniformis*, brewery waste for rhizospheric diazotrophs, and agroindustrial wastes are known to be suitable for *Cellulomonas* species [16]. Interestingly, biogas derived from the anaerobic digestion of sewage sludge and the discarded effluent has been used for SCP production using methanotrophic bacteria. The obtained SCP had a protein content of 41% (*w*/*w*) and the essential amino acids histidine, valine, phenylalanine, isoleucine, leucine, threonine, and lysine, making it suitable as a protein source for animal feed [115]. In fact, due to its high nucleic acid content (15–16%), which is degraded into uric acid, bacterial SCP might be more suitable for animal feed than human feed. While most vertebrate species can produce uricase enzymes, humans and other hominoids do not; therefore, uric acid is usually excreted in the urine. When uric acid excretion is impaired, leading to excessive accumulation in the body, hyperuricemia occurs. This metabolic disorder is linked to several health issues such as renal disorders, gout, diabetes, and cardiovascular diseases, among others [116,126]. Microalgae, however, contain lower contents of nucleic acids than yeast and bacteria, which often allows the direct use of their biomass. The amino acid and protein contents of microalgae SCP are strongly dependent on the species and cultivation conditions, with the latter varying between 30 and 80% (*w*/*w*) [116,117,118]. The incorporation of microalgae into food formulations is becoming well established, enhancing microalgae-SCP value chains where *Spirulina* is the most notable product. However, regular monitoring is essential to detect the presence of harmful constituents such as cyanotoxins, heavy metals, pesticides, or polycyclic aromatic hydrocarbons is advisable. Nonetheless, *Spirulina* is widely recognized for its health benefits and is promoted as a valuable source of dietary protein of high nutritional value. It has been shown to enhance the immune function and reduce the risk of cardiovascular disease, degenerative chronic disease, and cancer, among others [119,127]. In addition to its richness in vitamins, minerals and dietary fibers, microalgae are considered to meet the necessary requirements for human consumption by the World Health Organization (WHO). Despite their nutritional value, microalgae might carry harmful substances for human or animal consumption, including heavy metals, toxins, microbial or viral contaminations. Thus, processing microalgae biomass is a challenging task, currently in need of further development [120,121].

## 3. Challenges and Future Directions in SCP Production

### 3.1. Challenges and Limitations

It is increasingly evident that conventional agriculture alone may not be sufficient to meet future global food demands [87]. As the global population continues to grow and protein consumption increases, concerns about nutritional security are becoming more pressing. In context, there is an urgent need for alternative and more sustainable food production systems that ensure efficiency while minimizing environmental impact. The cultivation of edible biomass from unicellular microorganisms, known as SCP, presents a promising biotechnological solution. As discussed in this review, SCP offers a highly concentrated source of protein with a balanced profile of essential amino acids, and may provide additional health benefits, particularly related to cardiovascular and metabolic health, positioning it as a hopeful candidate to address global food challenges and improve health [105].

Despite the potential of SCP, several important challenges and limitations remain:Complexity of lignocellulosic biomass processing: The conversion of lignocellulosic biomass, a renewable and abundant feedstock, into fermentable substrates for SCP production involves multiple complex steps. These include size reduction and various physical, chemical, and biochemical pre-treatments aimed at breaking down plant cell wall components into fermentable sugars. Such processes often require specific conditions (e.g., pH, temperature and chemical concentrations) that differ from those used in traditional SCP production. Furthermore, these pre-treatments may generate inhibitory compounds that can negatively affect microbial fermentation, necessitating additional biomass conditioning steps [128].Nutrient limitations: Lignocellulosic substrates are deficient in nitrogen and other essential nutrients required for microbial growth, and therefore supplementation is often required [128].Enzymatic hydrolysis requirements: Even after pre-treatment, microorganisms are typically unable to directly metabolize complex polymers such as cellulose, hemicellulose, pectin and starch. These compounds must first be hydrolyzed into simpler sugars using enzymatic saccharification, requiring a suite of enzymes such as endo-glucanases, exo-glucanases, β-glucanases, and xylanases. These additional enzymatic steps significantly increase the cost of production when using agroforestry-derived biomass [128].Consumer acceptance: SCP products, particularly those intended for human consumption, often have undesirable flavors, odors, and colors. Additionally, obtaining pure SCP can be difficult due to residual lignocellulose particles that remain after fermentation. While purification techniques can be developed, they may increase the overall production costs [129].Safety concerns: Although most microorganisms used in SCP production are generally recognized as safe, contamination by pathogenic microbes can occur at various stages along the production and supply chain. This necessitates strict hygiene protocols and quality control standards to ensure product safety and protect public health. Moreover, the safety and sensory characteristics of SCP products depend on the microbial source. For instance, microalgae may pose a risk of toxin contamination; mycoproteins may trigger allergic reactions; and bacterial SCP requires careful strain selection to avoid pathogenic variants [116].Lack of industrial-scale validation: Current knowledge regarding the nutritional value of SCP is promising, but most evidence is based on laboratory-scale studies. Additional data from industrial-scale operations are needed to validate its safety, nutritional composition, and health impacts in real-world settings [130].Economic viability: The need for extensive pre-treatment, enzymatic hydrolysis, and purification steps increases the overall cost of SCP production, especially when using lignocellulosic biomass. These costs limit its competitiveness with conventional protein sources and represent a major barrier to commercial scalability [128,129].

### 3.2. Future Directions

The potential to generate high-value protein without compromising environmental integrity positions SCP as a promising solution to help address global food security and public health challenges. Moreover, SCP can be produced at any time because of its seasonal independence. Nevertheless, further research is essential to fully understand its health implications and facilitate its successful integration into the human diet [12,116].
Safety and regulation: Advances in microbial engineering and synthetic biology, particularly the development of microbial cell factories, have the potential to enhance SCP’s cost-efficiency, nutritional properties, and functional versatility, further improving its market competitiveness [131]. Adding to this, factors such as affordability, cultural barriers and relevant legislation need to be considered. Regarding the principles for food and feed safety, and the way specific co- and by-products might be used, the key regulatory frameworks in the EU have been defined by the General Food Law Regulation (EC) No 178/2002 and the Regulation (EU) No 68/2013. Approval processes usually involve a comprehensive safety assessment of allergens, toxicity, nutritional composition, and contaminants. Given this, SCP approval is subject to several regulations depending on its intended use, as feed and food legislation is determinant in shaping the development path of alternative protein sources. Therefore, it is essential to comply with regulations to allow for a safe, sustainable and transparent development of the global protein supply chain [4,6,14].Industrial investment and slow progress: Several companies worldwide are already investing in SCP technologies, gradually expanding their applications and commercial viability. However, despite its potential to help meet global protein demand, progress in SCP production has been relatively slow over the past six decades [87].Need for standardization: SCP remains at an early stage of development, and there is a pressing need to establish standardized methodologies that ensure consistent yields, quality, and safety. Standardization efforts will require close collaboration among academic researchers, industry stakeholders, and funding agencies [79].Importance of commercial scaling: Commercial expansion scaling is essential not only to meet immediate protein supply demands but also to advance the broader goals of the circular economy.Environmental advantages of lignocellulosic SCP: SCP produced from lignocellulosic biomass offers clear environmental advantages, such as reducing greenhouse gas emissions and minimizing land use. Beyond the microbial conversion of cellulose and hemicellulose into fermentable sugars for SCP production, future research should emphasize integrated biorefinery strategies that valorise all polymeric fractions present in lignocellulosic biomass. This includes not only the saccharification of structural polysaccharides (cellulose and hemicellulose), but also the conversion of residual hemicellulose and lignin into high-value polymeric materials. For instance, hemicellulose can be used to produce functional oligosaccharides and biodegradable films [132], while lignin can serve as a renewable polyol source in the synthesis of bio-based polyurethanes [133]. These valorisation pathways contribute to a circular bioeconomy by maximizing resource efficiency and minimizing waste [134].Product development and sensory improvement: To fully realize these benefits, increased focus must be placed on the development of SCP-based products, improving their sensory appeal, and evaluating their long-term health effects.Contribution to food system sustainability: Ultimately, SCP has the potential to promote both physical and economic access to nutritious food, contributing significantly to the global transition toward sustainable food systems.

## Figures and Tables

**Table 1 polymers-17-02251-t001:** Advantages and disadvantages of fermentation methods for SCP production.

Fermentation Methods	Advantages	Disadvantages
**Submerged (LSF)**	- Precise control of temperature, pH, nutrients, and oxygen; - Continuous process in closed bioreactors; - Easier biomass purification.	- High operational costs; - Requires rigorous substrate pre-treatment; - High oxygen demand; - Significant technical investment.
**Semi-solid**	- Lower cost; - Utilises agricultural residues; - Improves nutrient availability; - Allows better process control compared to SSF.	- Requires strict control of variables such as oxygen and inhibitory compounds; - Complex bioreactor design; - Needs agitation and heat removal.
**Solid-state (SSF)**	- High protein productivity, especially for fungi; - Low energy consumption; - Low capital investment; - Minimal waste generation.	- Limited nutrient diffusion; - More difficult control of parameters like temperature and moisture; - Requires intermittent aeration and agitation.

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
