# Peer review of "From Lignocellulosic Residues to Protein Sources: Insights into Biomass Pre-Treatments and Conversion"

_polymers, 2025, doi:10.3390/polym17162251_

Round 1
Reviewer 1 Report
Comments and Suggestions for Authors
Isabela V. dos Anjos et al have written the review entitled 'From Lignocellulosic Residues to Protein Sources: Insights into Biomass Pre-treatments and Conversion', which is a welcome contribution to this emerging field of valorization of lignocellulosic feedstock for value-added products. However, this needs "major revision" on the lines provided below, before it is ready to be accepted in the journal 'Polymers'.
- Please provide global protein market and also state what the issues are in the supply chain clearly?
- Why are you rushing to the SCP directly, you should talk about background of the major source of protein supplement such as whey and / or casein and then inroduce SCP.
- Pl ackenowledge the limitation of SCP in single statement.
- Authors stated that 'Fermentable sugars can be found in lignocellulosic biomass', this is very generic statement and authors must mention at least one or two sugars those are reported in high quantities in lignocellulosic biomass to support their claim.
- Under 2.1 Lignocellulosic biomass, please provide estimates of how much global production of Lignocellulosic biomass on year basis? from various sources e.g. forest, agriculture residue etc.
- Under 2.2.1. Lignocellulose pre-treatments, authors have described various methods available, further they have also acknowledged that , no single pre-treatment method can efficiently convert all types into free sugars; still this reviewer want them to clearly give recommendations of whihc method or compbnation or sequence of methods one needs to follow - while considering sugars useful for SCP.
- Authors have mentioned "is judged not only by its effectiveness but also by the overall energy consumption and operational costs...", please elaborate more on energy consumption and operational costs, as this is a review paper.
- It has been mentioned 'Additionally, environmental protection and safety concerns must also be considered' can you please specify what are the environmental protection and safety concerns, this is necessary for readers to understand your point-of-view more effectively.
- Under 2.2.2. SCP production methods, this section is more important, but unfortunately, wordy, this reviewer suggest in incorporate a table describing advantages and disadvantages of various methods discussed.
- Under 2.3 Microorganisms used in Single-Cell Protein production, please talk about molecular identification of at least potential strains.
- Is there any speccies - whose whole genome has been sequences? or there has been transciptomics studies carried out? if so please include this information.
- As stated before, 2.4 Nutritional Benefits of SCP, this section is more important, but unfortunately, wordy, this reviewer suggest in incorporate a table describing nutritional composition reported so far in the litrature.
- 3 Future Directions and Challenges in SCP Production, this needs to be separated, please first include Challanges and then Future Directions. Please provide information bulleted points in future directions. Please mention more specifically, limitations and more importantly, aspects of economic viability under challenges.
- SCP comes under edible sector and then you also need to provide policy guidance through your review that can be implemented by various governments. Please elaborate more on this under Future Directions.
Author Response
We appreciate the reviewer's comments. Responses to the comments are included in the attached file.

Reviewer 2 Report
Comments and Suggestions for Authors
Reviewer comments
The following points may be considered before publication of the manuscript “From Lignocellulosic Residues to Protein Sources: Insights into Biomass Pre-treatments and Conversion”.
- The hypothesis is not clear in the abstract; what will this review highlight?
- Avoid the use of abbreviations in the abstract.
- Some part of the introduction lacks the latest citations/references that must be added. Moreover, the introduction fails to explain the actual problem the authors want to address and is therefore suggested to be improved.
- Lines 172-213, this paragraph is too lengthy and must be revised to keep the flow in a consistent way.
- The majority of the statements lack a clear citation. Who says what?
- Almost all the figures are adapted from published literature. It is recommended to either obtain and submit the necessary copyright permissions for each figure as supplementary files or to modify all the figures accordingly.

English can be improved.
Author Response

(The authors gave the same response as above.)

Reviewer 3 Report
Comments and Suggestions for Authors
This review addresses the timely conversion of lignocellulosic residues into single‑cell protein (SCP). It surveys recent progress in pretreatment methods, fermentation routes, microbial platforms, and market outlooks. While the breadth is impressive and relevant to the polymer and biorefinery communities, the discussion needs deeper critical analysis and clearer presentation to become a reliable reference.
- The manuscript lists many pretreatments but fails to compare them quantitatively. Please add a table or summary that contrasts sugar yields, energy demands, and inhibitor formation for the chief methods and highlights the most suitable options for different biomass types.
- Regulatory and safety aspects are only briefly noted. Include a concise review of EFSA and FDA guidance on novel proteins, permissible nucleic‑acid levels, mycotoxin limits, and currently approved production strains.
- The introduction and Section 2 repeatedly justify SCP demand, producing redundancy and long paragraphs. Consolidate repetitive statements and break paragraphs longer than fifteen lines to improve readability.
- Several references are inconsistent, and raw URLs appear within the text. Conduct a full reference‑manager check, enforce MDPI style, and relocate URLs to the reference list (for example some have doi starts with http)
- In future‑work remarks, link SCP production to polymer valorisation (e.g., hemicellulose‑based films or lignin‑derived polyols) to engage the *Polymers* readership.
Author Response

(The authors gave the same response as above.)

Round 2
Reviewer 1 Report
Comments and Suggestions for Authors
Authors have revised the manuscript satisfactorily and the same may be accepted for publication in its current form
Reviewer 3 Report
Comments and Suggestions for Authors
Author have addressed each point clearly, provided appropriate justifications, and improved the manuscript’s clarity and rigor. I have no further comments. I recommend acceptance as is.